# Evaluating Overall Performance in High-Level Dressage Horse–Rider Combinations by Comparing Measurements from Inertial Sensors with General Impression Scores Awarded by Judges

**DOI:** 10.3390/ani13152496

**Published:** 2023-08-02

**Authors:** Sarah Jane Hobbs, Filipe Manuel Serra Braganca, Marie Rhodin, Elin Hernlund, Mick Peterson, Hilary M. Clayton

**Affiliations:** 1Research Centre for Applied Sport, Physical Activity and Performance, University of Central Lancashire, Preston PR1 2HE, UK; 2Department of Clinical Sciences, Faculty of Veterinary Medicine, Utrecht University, Yalelaan 112–114, NL-3584 CM Utrecht, The Netherlands; f.m.serrabraganca@uu.nl; 3Department of Anatomy Physiology and Biochemistry, Swedish University of Agricultural Sciences, S-750 07 Uppsala, Sweden; marie.rhodin@slu.se (M.R.); elin.hernlund@slu.se (E.H.); 4Biosystems and Agricultural Engineering and UK Ag Equine Programs, University of Kentucky, Lexington, KY 40546, USA; mick.peterson@uky.edu; 5Department of Large Animal Clinical Sciences, Michigan State University, East Lansing, MI 48824, USA; claytonh@msu.edu

**Keywords:** equestrian sport, performance, posture, balance, rider, dressage

## Abstract

**Simple Summary:**

Dressage is an Olympic equestrian sport in which scores are awarded primarily for the technical correctness of a horse’s performance. This study focuses on a single score awarded after the completion of a performance called the general impression score that is based on the gaits of the horse, the position of the rider, the effectiveness of the rider’s aids, and the harmony between the horse and rider. Twenty dressage horses and their riders performed a pattern from which walk, trot and canter, and transitions from extended to collected trots were analyzed based on data from inertial measurement units that measured three-dimensional accelerations and rotations of the horse’s trunk, the rider’s pelvis, and the rider’s trunk. The selected movements were observed on video and scored by two or three high-ranking dressage judges. The judged scores were then compared with the data describing the movements of the horse’s trunk, the rider’s pelvis, and the rider’s trunk. The score for a horse’s gaits was most heavily influenced by stride frequency, with a slower frequency being favored. The judges’ scores for posture, effectiveness of aids, and harmony with the horse were most strongly influenced by the asymmetries in a rider’s trunk movements, such that higher scores were associated with fewer rider asymmetries.

**Abstract:**

In the sport of dressage, one or more judges score the combined performance of a horse and rider with an emphasis on the technical correctness of the movements performed. At the end of the test, a single score is awarded for the ‘general impression’, which considers the overall performance of the horse and rider as a team. This study explored original measures that contributed to the general impression score in a group of 20 horse–rider combinations. Horses and riders were equipped with inertial measurement units (200 Hz) to represent the angular motion of a horse’s back and the motions of a rider’s pelvis and trunk. Each combination performed a standard dressage test that was recorded to video. Sections of the video were identified for straight-line movements. The videos were analyzed by two or three judges. Four components were scored separately: gaits of the horse, rider posture, effectiveness of aids, and harmony with the horse. The main contributor to the score for gaits was stride frequency (R = −0.252, *p* = 0.015), with a slower frequency being preferred. Higher rider component scores were associated with more symmetrical transverse-plane trunk motion, indicating that this original measure is the most useful predictor of rider performance.

## 1. Introduction

Dressage is an Olympic sport based on collaboration between a rider and a horse. The relationship is complex, with both partners being responsible for distinct roles in the performance. The rider learns a test pattern consisting of a series of movements that includes geometric figures performed at various gaits and speeds. The rider is responsible for communicating the requirements of the pattern to the horse nonverbally using almost imperceptible signals called ‘aids’. During training, the horse learns to interpret and respond willingly to subtle aids. A well-trained horse performs the movements willingly without confusion or resistance. As the level of skill increases, horses and riders compete through a sequence of competition levels that requires improved performance of the basic skills combined with the addition of more difficult movements. National federations develop tests up to a medium level of difficulty, and the FEI develops a series of tests of progressively increasing difficulty. These are, in order of increasing degree of difficulty, the Prix St. Georges (PSG), Intermediate (Int I, Int A, Int B, and Int 2), Grand Prix (GP) and Grand Prix Special (GPS). The small tour consists of the PSG and Int I, and the big tour consists of Int 2 and the GP.

Each movement that a horse and rider perform receives a score from one or more judges based on directives that describe observable criteria [1]. The performance of the rider, per se, is scored by the judge(s) after the test is completed in a single score called ‘general impression’ that is based on the harmonious presentation of the rider/horse combination, the rider’s position and seat, and the discreet and effective influence of the aids [2]. It is awarded up to a maximum of ten points and has a coefficient of two, so effectively, the general impression score receives up to 20 points.

Ideally, a dressage rider’s posture is based on the vertical alignment of the ear, shoulder, hip, and heel in the lateral view and a rider’s vertical alignment over the midline of the horse in the frontal view [3]. Several studies have reported the static and dynamic posture and motion patterns of riders at various gaits without consensus and sometimes in contradiction to each other [4,5,6,7,8,9,10]. The discrepancies may be partially explained by inexperienced riders having inferior strength or coordination that interferes not only with their ability to move the body segments independently, but also with controlling their motion under the influence of perturbations from the horse. For example, larger ranges of trunk motion in sitting trot and canter were found in beginner riders [9], which indicated the inferior use of erector spinae and rectus abdominis to stabilize the trunk [11]. While inexperienced riders may have larger ranges of uncontrolled rotation, experienced riders may use active pelvic rotation to affect the movements of a horse. For example, a more caudal rotation of the pelvis and greater flexion of a rider’s lumbar spine are used by experienced riders to collect the gaits [12,13].

Rider posture has been measured using video-based methods, motion-capture systems, and more recently with inertial measurement units (IMUs). IMUs use accelerometers and a gyroscope to provide data from which segmental orientations and displacements can be calculated [14]. Different IMU systems have been validated for equine use [14,15]. They have been used extensively in biomechanical studies of horses, particularly for detecting the vertical motion asymmetries of axial midline markers in relation to lameness [16,17,18] and for studying coordination between a horse and rider [10,19]. The benefits of using IMUs compared with video and motion-capture systems include simplicity of setup and the ability to record a large number of sequential strides.

Studies linking rider skills to overall horse–rider performance using motion-capture methods have been investigated. One of the early studies in this area [20] compared a professional rider to a recreational rider riding the same 20 horses at sitting trot. The angle between lines connecting the rider’s head to the rider’s back and the rider’s back to the horse’s head was determined. The length and deviation of the resulting angular vector within the phase space was calculated, and its deviation, which represented motion pattern consistency, was used as a measure of harmony. The recreational rider had larger values than the professional rider, indicating a less consistent motion pattern. Another study used artificial neural networks to analyze the time-continuous pattern of rider–horse interactions with 14 horses. Comparisons were made when the horses trotted in hand, were ridden by a professional rider, and were ridden by a novice rider. Both rider-specific and horse-specific motion patterns were identified, with the professional rider showing higher adaptation to the horses’ movement patterns [19].

Ideally, a rider should sit symmetrically relative to a horse’s back. The position and movements of a rider’s center of pressure (COP) can be tracked using an electronic saddle pressure mat. Each gait has a characteristic pattern of COP motion on a horse’s back [21]. Horses that are lame or markedly asymmetrical for other reasons may cause the saddle to slip predominantly to one side [22], making it impossible for a rider to stay centered. The laterality of riders has been studied in terms of left:right ratios of weight distribution [23], rein tension [24], and body posture [25]. However, mounted measurements of rider symmetry may be confounded by any side preferences shown by a horse. Many riders also suffer from chronic injuries that affect their weight distribution, posture, and balance [10].

Taken together, these studies illustrate the difficulty in identifying rider performance that can clearly indicate a higher level of skill. For a sport involving two somewhat independent athletes performing a complicated pattern of movements, the task of establishing key performance determinants is challenging. Holistically, a rider must regulate trunk balance in accordance not only with the gaits and movements being performed, but also maintain the stability of their head in space, as the head houses the visual and vestibular systems that are important for balance control [4,26]. The strategy used by a rider to achieve this may be gait- and horse-dependent [5,8,27], but it must maintain an impression of harmony and must not interrupt the rhythm of the horse [27,28]. As such, dynamic postural control of a rider’s trunk through regulated, coordinated trunk-to-pelvis rotational motion, however individualized the pattern, is expected to be an attribute of rider skill.

Thus, the aim is to investigate holistic, objective measures of trunk and pelvis posture, stability, and coordination that can be used to quantify overall rider performance in dressage. We address this by comparing subjective scores of experienced judges with objective data from IMUs to determine which qualities are rewarded by the judges in the collective marks that form the ‘general impression’ score. It is hypothesized that visual information contributing to the judges’ general impression scores for high-level riders include the impression of where the trunk is in space, how well the motion of the pelvis follows the motion of the horse, and the quality of the movement of the horse below the rider. As such, it is expected that measurements related to these impressions are predictive of judged scores for general impression.

## 2. Materials and Methods

### 2.1. Participants

Twenty-one dressage horses were evaluated trotting in straight lines on a firm, level surface by two experienced lameness clinicians (authors MR and EH) prior to acceptance into the study. One horse was excluded due to losing a shoe on the day of testing. The regular riders of the accepted horses were informed of the requirements of the study and the procedures to be used. The profiles of the horses and riders are shown in Table 1. Thirteen horse–rider combinations were unique, but three riders rode two different horses. One GP rider rode two different GP horses, and two GP riders rode one GP horse and one Small Tour horse. Written informed consent was provided by the riders prior to commencing the study.

Inertial measurement units (IMUs, 200 Hz) were attached mid-dorsally to the riders’ sacra, the riders’ thoraxes at the level of T8, the horses’ polls between the tubera sacrale, and over the horses’ lumbar regions behind the saddle. Sensors were aligned with the inferior-superior (X) and mediolateral (Y) axes of each segment for the riders and the craniocaudal (X) and mediolateral (Y) axes of the horses’ trunk segments. In addition, sensors were attached to all hooves for stride segmentation. Video data were recorded during the motion trials at 25 Hz with a camera positioned 2–3 m behind A or C. The camera was synchronized with the IMUs prior to starting the tests by a researcher swinging their arm through a large range of motion before tapping a sacrum sensor. The arm motion was easily detected on the video recording and facilitated identification of contact with the sensor. The horses stood for 10 s during the synchronization.

### 2.2. Procedures

After being tacked up but prior to being mounted by the riders, the horses were trotted in hand on a firm, level surface while recording data from the IMUs. MinDiff and MaxDiff values, which represented the mean differences between the minimal and maximal heights, respectively, of the markers on the poll and sacrum were checked after the trot in hand, and horses with asymmetry values exceeding previously established values for the IMU system used (poll MinDiff: 13 mm, MaxDiff: 12 mm; sacrum MinDiff: 5 mm, MaxDiff: 4 mm) [29] were removed from the study.

Riders wore riding breeches, boots, a tightly fitted shirt, and a helmet. After warming up, each horse–rider dyad performed a dressage test (Appendix A) written for this study (HMC) to evaluate specific aspects of horse and rider performance. The tests were performed in a 20 m × 60 m arena (Figure 1) with sand and geotextile/fiber footing. The gaits analyzed were collected walk on a straight line, collected trot and collected canter on straight lines on the left and right reins, and transitions from extended to collected trots on straight lines on left and right reins. During postprocessing, the gaits and movements of interest were identified via visual examination of the video footage, and the video time codes were used to select the time-synchronized segments of IMU data for analysis. For the steady-state gaits, nine strides were selected from the mid-section of the IMU data string to omit turning effects.

For transitions, the time codes from the video data were used to identify the transition strides. The location of each gait or movement is shown in Figure 1.

A pool of 16 dressage judges who were certified at the senior (S) level by the United States Dressage Federation or certified by the Fédération Equestre Internationale were recruited. The judges were randomly assigned to judge the 20 dressage tests, with different combinations of judges used for each test. This method was chosen to distribute high- and low-scoring judges across the different horse–rider combinations. Each judge scored the performances of the horses and riders for each movement on a scale from 0 to 10 using the same video segments from which the IMU data were extracted. Two or three judges scored each horse–rider combination, and their scores were averaged per gait/movement prior to statistical analysis. After completing the test, the riders were judged on three rider scores that represented the components of the general impression score at the end of the FEI test:the rider’s position and seat;the use of discreet and effective aids to influence the horse;the overall impression of harmony between the rider and the horse.

Contact and lift off for each hoof were identified using the limb IMU data, as previously described [30]. IMU signal components from the sensors together with contact and lift off signals were extracted from Matlab as one file for each horse–rider combination and imported into Visual 3D for further processing. A critically damped 4th-order low-pass Butterworth filter was used to remove high-frequency noise at 30 Hz for acceleration signals and 10 Hz for gyroscope signals. All signals were then high-pass-filtered at 0.1 Hz to remove systematic drift [14,31].

Stride frequency was determined from the sensors on the hooves as the inverse of the time elapsing between the successive contact of the inside hind limb. The mean stride frequency for each gait and movement section was extracted prior to further processing. Acceleration signal components were integrated to obtain velocity, centered around zero by removing the mean, and then integrated to obtain displacement. Each waveform was high-pass-filtered at the walk stride frequency of the horse–rider combination to remove cyclical drift (0.77 to 0.96 Hz), following the method described by [14]. After initial filtering, gyroscope signal components were integrated to obtain angular data, but no further filtering was performed.

### 2.3. Performance Variables

As stride frequency is associated with horse performance [32], this variable was retained as a performance variable for inclusion in the analysis. For each horse–rider combination, the rider performance measurements described below were calculated from the processed IMU waveforms.

#### 2.3.1. Rider Stability

The stability of the riders’ trunk and pelvis segments was measured via comparing each segment’s acceleration to the acceleration of the horse’s back. Stability was evaluated using the filtered acceleration data. A constant of 300 ms^−2^ was added to the X,Y,Z acceleration signals to ensure that all values were positive. The resultant 3D acceleration of a horse’s back, a rider’s pelvis, and a rider’s trunk were calculated at each time point, and the mean value was subtracted to center the data around zero. For each segment (rider trunk, rider pelvis, and horse trunk), the root mean square (RMS) acceleration of the sequence was calculated, which is commonly used as a measure of postural stability [33], and exported to Excel. The difference between the horseback RMS acceleration and rider segment RMS acceleration was calculated (horse back − rider segment) and expressed as the ratio of horse back RMS acceleration. Positive values indicated that a horse’s back moved more than a rider’s segment (trunk or pelvis) and vice versa.

#### 2.3.2. Rider Coordination

A rider follows the movements of a horse using pitch, roll, and yaw rotations of the trunk and pelvis. The coordination strategies used to cope with different gaits and speeds are highly individualized and horse-dependent. To compare rider coordination strategies on different horses, rider coordination variability was assessed by evaluating the detrended angular impulse of 3D resultant trunk-to-pelvis rotation. By summing the angular change over the stride time, the ‘persistence’ of rotation between segments could be measured as an indicator of the similarity in coordination of the resultant rotation in one stride compared to others. The method was adapted from detrended fluctuation analysis methods [34] but used an alternative outcome variable to alpha due to the length of available straight-line motion that could be included in the analysis.

The rotation of the trunk relative to the pelvis was calculated for each orthogonal rotation and the 3D resultant determined, as outlined previously for horse back 3D rotation. Each gait sequence was separated into strides (9 strides) and time-normalized to 101 points. These data were exported to Excel. The ensemble mean of the 909 points was calculated, and this value was subtracted from each individual data point to center the data set around zero. For each stride, the 3D resultant rotations between the trunk and pelvis were summed to obtain the angular impulse. Each individual trial was then detrended via subtracting the linear trend from the waveform. Finally, the absolute difference between each data point and their averages across the nine trials were calculated. To compare between riders, the average deviation in detrended angular impulse over the nine trials was calculated (DVar), which was the summed absolute difference of all the data points divided by the number of data points [35,36]. An example of the results of performing these calculations is shown in Figure 2.

#### 2.3.3. Rider Transverse Dynamic Symmetry

Left–right dynamic symmetry was evaluated using mediolateral (Y) and anterior-posterior (Z) displacement of the trunk and pelvis segments in their respective segment coordinate systems (SCSs). For each gait sequence, the average motion pattern over nine strides was plotted in Visual 3D, exported as a time-normalized ASCII file, and imported into Excel. Using the first mediolateral crossing point after the inside hind foot strike, the Y and Z average displacements were spilt into two halves, each representing one half of the time-normalized data. The right half of the data was then folded over the left by multiplying the right Y axis by minus 1. The Z alignment depended on the pattern. If the two halves crossed in the center of the pattern, then the anterior-posterior pattern was not transposed. If the motion was circular, then the Z axis was reversed. Plots of each trunk and pelvis pattern for each gait and each horse–rider combination were produced to ensure that the data were transposed correctly. The average position between the two halves was calculated for each corresponding time point, as illustrated in Figure 3. Following the method illustrated by [37] to describe natural asymmetry in plant species, transverse dynamic symmetry (TDS) was determined by calculating the sum of the root mean square deviation for the dynamic motion pattern, as described below.
$$Transverse\;Dynamic\;Symmetry=\underset{i}{\sum }\sqrt{{\left(Y_{i}-Y_{ave}\right)}^{2}+{\left(Z_{i}-Z_{ave}\right)}^{2}}$$
where *i* = an individual timepoint;

*Y_i_* = the mediolateral position (left/right) at that timepoint;*Z_i_* = the anterior-posterior position (left/right) at that timepoint;*Y_ave_* and *Z_ave_* = the calculated average positions at that timepoint.

**Figure 3 animals-13-02496-f003:**
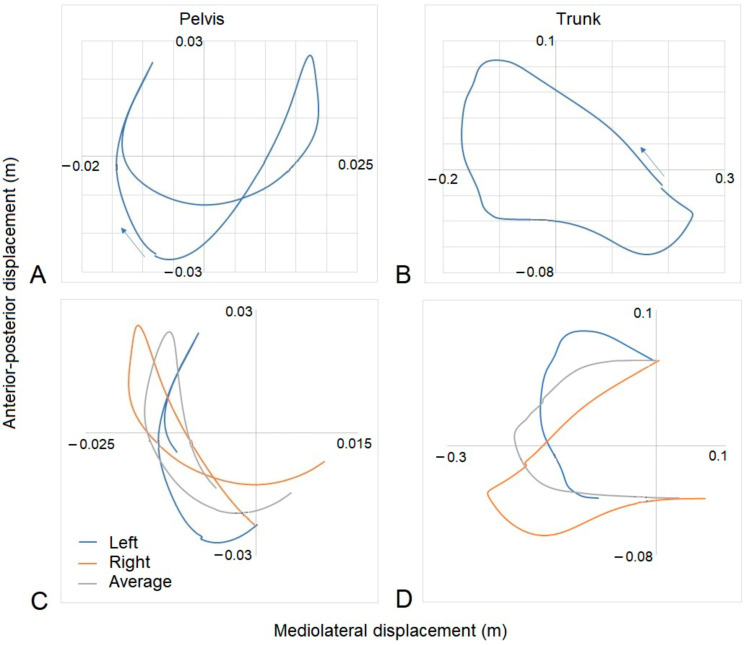
Examples of the plots developed to determine transverse dynamic symmetry for right trot from one GP rider. (**A**) Pelvic and (**B**) trunk mean transverse motions showing the directions of motions with arrows beginning at inside hind limb foot strike. The plots are viewed as if looking down on the rider from above, so anterior displacement is from 0,0 upward, posterior downward, right to right, and left to left. (**C**) Pelvic and (**D**) trunk left and right halves and average dynamic motion patterns from which dynamic symmetry was calculated. Note that the pelvic motion was not transposed, whereas the trunk motion was transposed.

### 2.4. Statistical Analysis

Two analyses were performed to investigate skill level, one according to rider skill level and one according to horse skill level, where level was categorized as (1) GP or (2) Small Tour. Descriptive statistics were produced for performance variables and judged gait scores separated by gait/movement, rider level, and horse level. Differences in performance measurements and judged scores were analyzed using a within-subjects linear mixed effects model with gait/movement as repeated measures, rider level and horse level as fixed factors, and using the restricted maximum likelihood method. For general impression judged scores, independent sample *t*-tests were performed to analyze the differences in scores between rider and horse levels separately, and equality of variance was assessed using Levene’s test.

Stepwise linear regression was then used to identify performance measurements that were predictors of judged gait and collective scores. The assumption of independent errors and collinearity diagnostics were investigated using the Durbin–Watson statistic and the variance inflation factor (VIF), respectively. Significance was set to *p* < 0.05. All statistical analyses were performed with SPSS.

## 3. Results

Descriptive statistics for performance measures and judged gait scores are presented in Table 2 and Table 3 for rider-level results and horse-level results, respectively, with significant differences (*p* < 0.05) between groups identified for each measure. For general impression scores, significant differences (*p* < 0.05) were found for rider level but not horse level (see Table 4). One of the 20 accepted horse–rider combinations was not included in the analysis, as the quality of the video footage was not deemed sufficient. From the 19 horse–rider combinations that were included, for one horse–rider combination, trot data were missing from the video and, for another horse–rider combination, left canter was not performed. As the transitions were performed over one or two strides, insufficient data were available to calculate rider coordination variability for these movements. In relation to horse–rider combinations, one Small Tour rider rode a GP horse and seven GP riders rode Small Tour horses. For all other horse–rider combinations, horses and riders were at the same level.

When comparing between rider levels, no significant differences (*p* < 0.05) were found between GP riders (n = 14) and Small Tour riders (n = 5) for performance variables or dressage scores.

For horse level, the groups were GP (n = 8) and Small Tour (n = 11). When comparing between horse levels, no significant differences (*p* < 0.05) were found between GP horses and Small Tour horses for performance variables or dressage scores.

When performing the stepwise linear regression to identify performance measures that could significantly (*p* < 0.05) predict judged scores for gaits/movements and judged collective scores, transverse dynamic symmetry of the trunk (TDS trunk) was identified consistently when all gaits and movements were included in the model. Stride frequency predicted 6% of the gait/movement scores, and with TDS trunk included in the model, an additional 7% of the gait/movement scores could be predicted. TDS trunk predicted from 14 to 18% of the collective scores, which increased to 21 and 24% for aids and position, respectively, when pelvis stability was also included in the model. The results of the regression analysis are provided in Table 5.

Examples of average transverse-plane motions of the trunk, pelvis, and horse’s back from two horse–rider combinations are shown in Figure 4. The rider on the left side had competed up to the PSG at the time of testing and was riding a Small Tour horse with an average gait score of 6.3 and a general impression score of 6.6. The transverse motion of the horse’s back was small, and the motion of the rider’s pelvis was larger and relatively symmetric in walk, asymmetric in canter, and minimal in trot. The rider’s trunk described a much larger, asymmetric pattern, with a tendency of moving anteriorly rather than laterally to the left. This pattern is typical of an inverted pendulum model, with more trunk motion than pelvis and horse back motion. The rider on the right was a higher-scoring GP rider riding a GP horse with average scores for gait of 7.7 and for general impression of 7.6. This horse had a larger motion pattern, particularly in walk and canter. The rider’s pelvis described a similar motion pattern in canter to the lower-level rider but had a less distinct pattern mediolaterally at walk. The rider’s trunk motion was minimal and more symmetrical; hence, their results for TDS trunk were small. For this rider, the trunk appeared to be a more consistent vertical reference from which motion below it occurred.

A stepwise linear regression was also performed for individual gaits and movements to assess whether performance variables were specific predictors of gait scores. Transverse dynamic symmetry of the pelvis (TDS pelvis) was a significant predictor for walk (r = −0.519, *p* = 0.023). Higher gait scores were associated with more symmetrical motion of the pelvis and, hence, lower scores for TDS pelvis, which predicted 27% of the judged scores. No other relationships were found.

The Durbin–Watson statistic was between 1.811 and 2.478 for all of the regression analyses, indicating that independent errors were not evident (Table 5). The variance inflation factor (VIF) values, which detected the severity of multicollinearity, for each model and the excluded variables were from 1 to 1.723, indicating that collinearity was not evident in the analysis.

## 4. Discussion

This study investigated measurements that could be used to objectively evaluate skill in high-level dressage riders. Drawing upon previous work, holistic measures of rider performance were developed to quantify visual information that was hypothesized to contribute to the general impression score, and these measurements were evaluated against judged dressage scores. We hypothesized that the impression of where the trunk was in space would contribute to the judges’ general impression scores. Overall, transverse dynamic symmetry of the trunk (TDS trunk) was found to be the strongest predictor of the general impression scores and contributed to 7% of the gait scores. We hypothesized that the ability of a rider to follow the motion of a horse with their pelvis would also contribute to the judges’ general impression scores. Pelvis stability made a small contribution to the general impression scores, and for walk, transverse dynamic symmetry of the pelvis (TDS pelvis) contributed to more than one-quarter of the walk gait scores. The quality of the movements of the horses was quantified using the stride frequency for each gait/movement, and indeed, this variable was the strongest predictor of gait scores. Skill was also evaluated by the competition levels of the riders and horses in two separate analyses. No significant differences (*p* < 0.05) were found between either rider levels or horse levels for measurements or gait scores, but significant differences (*p* < 0.05) were found between rider levels for the general impression scores of harmony and aids.

In previous studies, trunk rotation has been measured and has been found to differ between riders and gaits [13,38]. Pitching movements of a rider’s body segments are not only influenced by the rotations of a horse’s back, but also by longitudinal forces. The synchronized limb movements and forces in trot produce a large decelerating force in early diagonal stance and a large propulsive force in late diagonal stance. Experienced riders anticipate the resulting perturbations and counteract their effects with a feed-forward response that involves pitching the trunk caudally in early stance and cranially in late stance [39]. Co-activation of a rider’s antagonistic core muscles makes the spine system more robust to external perturbations [40], and the greater strength of the anteroposterior core musculature resists the pitching accelerations of a rider’s trunk due to the effects of braking and propulsive longitudinal ground reaction forces when the degree of co-activation is appropriate to the task at hand [41]. Since the core muscles are less effective in stabilizing the trunk mediolaterally, the coupling between rider and horse is weaker in this direction [5,42], and it is more difficult for a rider to maintain stability. In previous studies, due to the smaller accelerations and decelerations of a horse’s trunk in walk, the rotation of a rider’s upper body relative to the pelvis varied between riders. In some riders, the trunk had a biphasic pattern during each hind limb stance, pitching cranially at the times of maximal and minimal pelvic pitch. In other riders, no distinct pattern was recognized due to the large variability between strides [38]. A further example of rider variability comes from a study of six experienced riders in sitting trot. Five riders used a similar pattern of pitch rotations of the trunk and pelvis to absorb the oscillations of the horses, while one rider used a very different strategy compared to the rest of the group [39]. As such, identifying skill from phasic rotations around specific axes and linking the measurements to judges’ observations adds many layers of complexity. In this study, we also found that coordination variability between 3D motion of the pelvis and trunk (DVar) was not a feature of higher-level performance due to between-rider differences.

Transverse dynamic symmetry measures for the trunk and pelvis were developed for this study to quantify holistic, asymmetric motion patterns in riders. By comparing each half of the motion pattern at each normalized time point, the measure accounted for both spatial and temporal left-to-right differences, but the measurement also increased when pitching motion was evident. To maintain the trunk and, therefore, head in-space stability, the trunk is often modeled as an inverted pendulum [26]. For a dressage rider, observable rotational motion of the trunk is not a desired outcome. Instead, using the trunk as a vertical reference from which the pendulum motion of the pelvis and horse below move smoothly and in harmony is considered preferable. Stakeholders previously described how high scores in dressage competitions included the ability to maintain dynamic postural control in order to absorb the movements of a horse and provide a stable platform for applying aids with the seat, legs, and hands that directly influences the quality and accuracy of a horse’s movements [43].

The present study expanded on our previous work by showing that TDS trunk was the most prominent predictor of the general impression scores from the gaits and movements assessed and was a contributor to the prediction of gait scores. Its value may be related to the fact that it is influenced by all the movements below it together with capturing a rider’s perception and control of trunk movements. In other work, researchers have described how experts use a hierarchical model to observe dance movements, focusing on larger ‘chunks’ or ‘phrases’ that are used to assess complex dance movements without the need to retain the specific details [44,45]. In another study, expert gymnastic judges were found to view 27% fewer fixations of performances when focusing more on the upper body compared to novice judges who focused more on lower limbs [46]. Similar descriptions of expert perception-of-motion aesthetics in free-running skills [47], as well as amongst other aesthetically judged sports [44], have been reported. From our results, we surmise that TDS trunk captured a ‘chunk’ of the motion aesthetics that were observed by judges and formed part of the judges’ assessments of rider performance. Further work is needed to confirm this, as one thesis reported more fixations on the lower legs of riders [48]. Another reported a lower duration of fixation time in an expert compared to a novice judge, concluding that an expert could process visual information more effectively [49].

Walk has been described as the most difficult gait to ride, and TDS pelvis predicted more than one-quarter of the gait scores for walk. During the first half of hind limb stance, a horse’s trunk pitches nose-down and rolls and yaws toward the hind limb as it accepts weight [38]. A rider’s pelvis pitches caudally in the opposite direction of a horse’s trunk while simultaneously rolling and yawing in the same direction as a horse’s trunk. Less-skilled riders move their pelvises ahead of a horse, disrupting the horses’ natural motion [27], which is expected to influence the perception of gait quality. The relationship between TDS pelvis and judges’ gait scores may reflect the fact that a horse’s back undergoes considerable bending motion in walk, so even small spatial and temporal asymmetries between horse back and pelvic motion can affect the quality of a walk.

The addition of pelvis stability in the general impression scores was a negative relationship, so smaller 3D accelerations of a rider’s pelvis compared to a horse’s trunk were associated with higher scores. This could suggest better coupling with a horse, as smaller differences were indicative of more stability in the saddle and larger differences, overall, were identified in canter. However, pelvis stability contributed to the scores for harmony and aids. In another study evaluating the functional abilities of riders, we found that riders who caused minimal disruption to a horse during the performance of a gait or movement received higher scores [50]. A rider’s pelvis is responsible for following the motion of a horse’s back at all gaits through supple movements of the lumbosacral and hip joints, which explained its relationship with the score for harmony. Another function of pelvic rotations and the associated muscular forces is to give aids that regulate the degree of collection [12,13] and, thus, they were considered to contribute to the score for aids.

Stride frequency was selected as an indicator of gait quality, which is in agreement with other studies that have related lower stride frequencies to higher scores in dressage [32]. Stride frequency depends on stance duration and swing duration, which vary with factors such as limb length and speed of locomotion. Speed of locomotion had different relationships with stride length, which increased linearly with speed, and stride duration, which increased nonlinearly and more slowly [51]. Stance duration tends to be preserved to allow time for a limb to develop the necessary propulsive forces without generating excessively high force peaks. Also, longer limbs allow a body to rotate further forward over a grounded hoof during the stance phase [52], which requires more time to cover the longer distance [53]. It is, therefore, unsurprising that stride frequency was a predictor of gait scores.

Comparisons between horse and rider levels did not reveal significant differences in measurements or judged scores. It has been shown that the gait kinematics of four-month-old warmblood foals could be used to predict many of the gait variables of mature horses, in spite of the effects of growth and maturation. This was described as the “kinematic fingerprint” [52]. Training may refine locomotor patterns through improvements in balance, skill, and coordination [54], but in spite of a rider’s efforts to improve gaits with training, the underlying characteristics are retained. As all horses were selected for having gaits that were appropriate for the sport of dressage, this may explain why no difference was found for horse level.

Although the horses in the study described here performed only the basic gaits, it seems counter-intuitive that riders of high-level horses showed larger values that were closer to significance for TDS trunk than those riding lower-level horses. The most likely explanation is related to the fact that higher-level riders who have longer riding careers are more likely to show pelvic asymmetry during sitting and reduced ROM in lateral bending to the left [10]. Riders do not always sit in vertical alignment over the midline of a horse with their weight evenly distributed over the left and right tuber ischii [55]. A study evaluated riders who habitually sat with the pelvis rolled to one side, usually the left, which was associated with more force on the right side of a horse’s back [55]. Another possibility is the transmission to a rider of subtle asymmetries in a horse, possibly associated with mild chronic injuries or inherent side preferences that did not reach the threshold for visual detection. Finally, judged scores were also not different between horse or rider levels. As all horse–rider combinations regularly performed the gaits and movements studied during training and competition, their performances on the day were more important than their current competition level.

The current study was performed in the field using horse–rider combinations that were actively competing in high-level dressage competitions. A limitation in the design is that some of the GP riders were riding less-experienced horses, but since this is a common occurrence in dressage competitions, it was not regarded as a reason to eliminate these combinations. The numbers of riders and horses competing at each level to perform the statistical analyses were also not balanced. Although a linear mixed model is robust, as assumptions are related to the distribution of residuals, the numbers of riders in each group may have influenced the statistical outcomes when comparing between levels. A smaller number of steady-state gaits and movements was used to investigate general impression scores, which may not fully represent the judges’ overall assessments of the general impression scores for an entire test. Therefore, these performance measurements predicted only a proportion of the judged scores, and further work to explore turning, lateral, and higher-level movements and gaits is needed.

## 5. Conclusions

We concluded that the scores for the horses’ gaits were higher in horses with slow stride frequencies ridden by riders with more aligned, symmetrical trunks. For the judges’ general impression scores, transverse dynamic symmetry of the trunk was the main contributing variable to the riders’ position, harmony with the horse, and effectiveness of aids. The importance of this variable in the performance of high-level dressage riders has not been reported previously. Pelvis stability, which is well-recognized as a determinant of performance, contributed to scores for the riders’ position and effectiveness of aids.

Our findings support the old equestrian saying: the rider’s pelvis belongs to the horse, but the shoulders belong to the rider.

## Figures and Tables

**Figure 1 animals-13-02496-f001:**
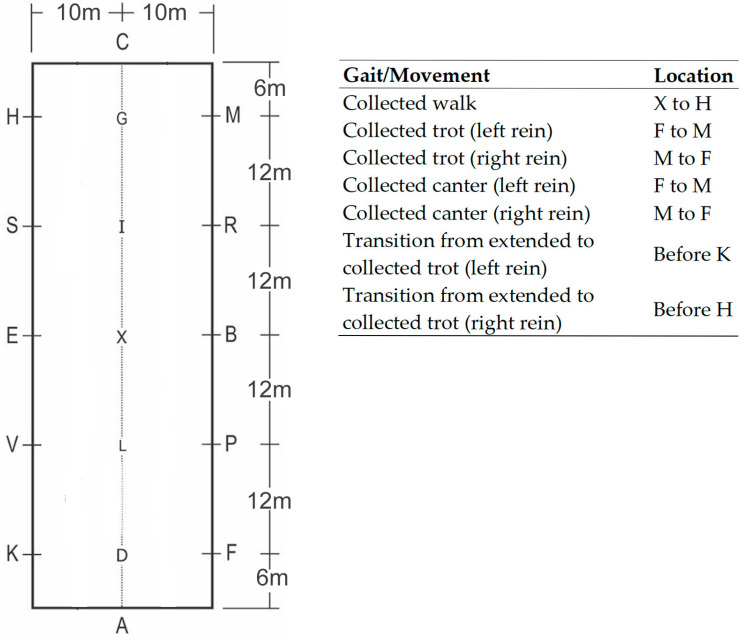
On the left is a diagram showing the dimensions and markers around the dressage arena. On the right are the dressage movements analyzed in this study and their locations in the arena.

**Figure 2 animals-13-02496-f002:**
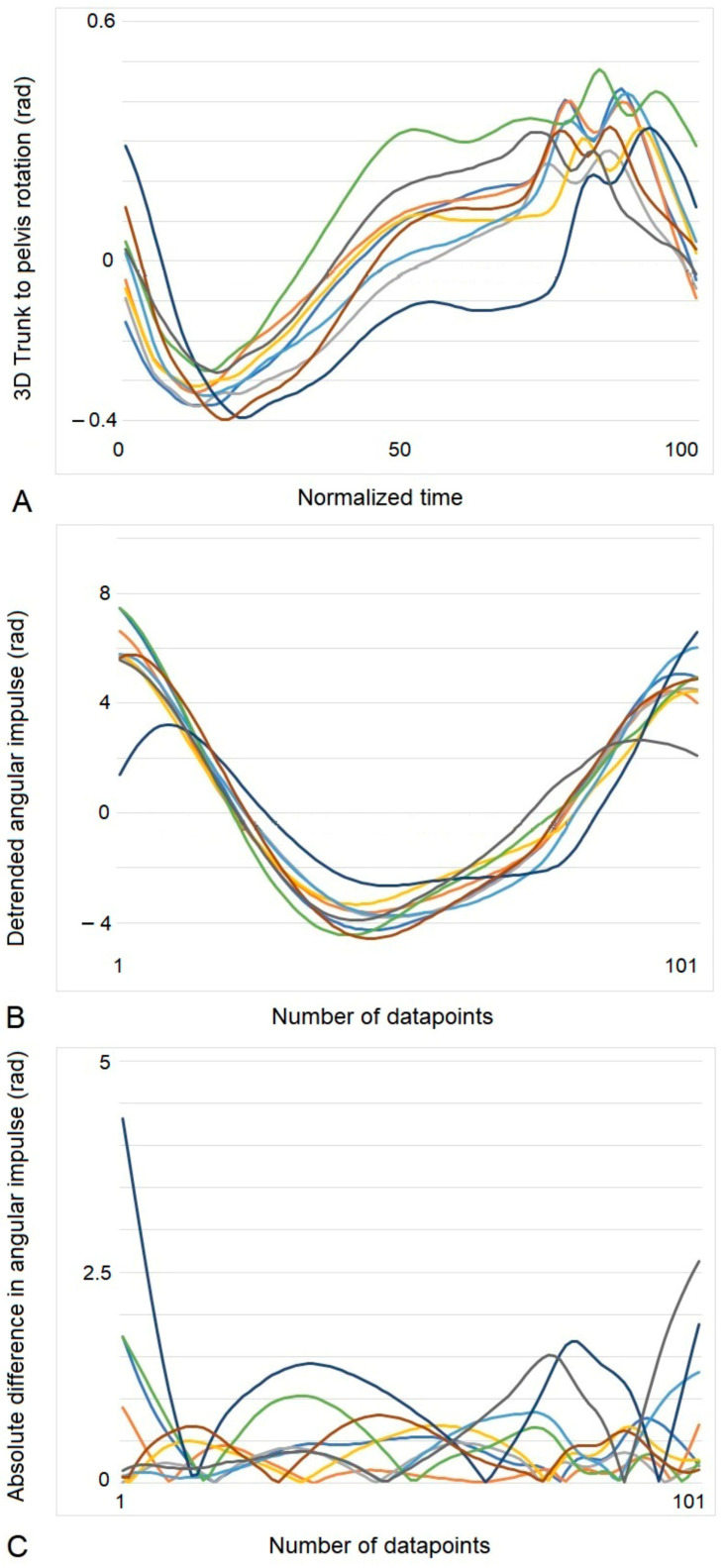
Results of data-processing steps to obtain rider coordination variability using data from one rider at walk. Colored lines represent the nine individual strides. (**A**) 3D trunk-to-pelvis rotation (rad). (**B**) Results after detrending angular impulse (rad). (**C**) Results following calculation of the absolute difference between the average angular impulse and the individual stride angular impulse at each data point. Rider coordination variability (DVar) was calculated by summing all of the data points in (**C**) and dividing by the number of points (909 points). Stride started at contact of the inside hind limb. Each stride is represented by a different color.

**Figure 4 animals-13-02496-f004:**
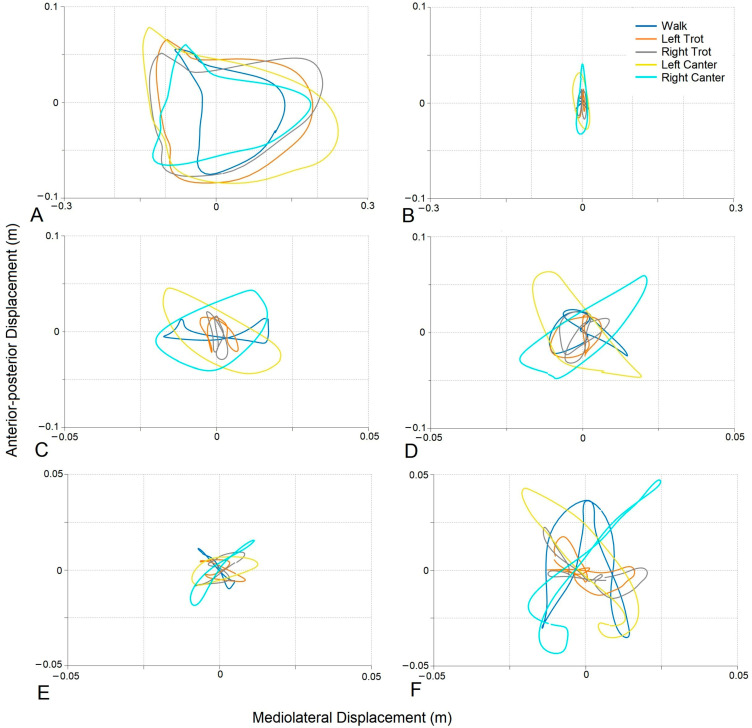
Examples of average transverse-plane motions (m) on the left from a Small Tour rider riding a Small Tour horse (average scores: gait = 6.3; general impression = 6.6) (**A**,**C**,**E**) and, on the right, from a higher-scoring GP rider riding a GP horse (average scores: gait = 7.7; general impression = 7.6) (**B**,**D**,**F**) over 9 strides of steady-state gait in a straight line. (**A**,**B**) Trunk displacement. (**C**,**D**) Pelvis displacement. (**E**,**F**) Horse back displacement. Walk = blue; left trot = orange; right trot = grey; left canter = yellow; right canter = cyan.

**Table 1 animals-13-02496-t001:** Participant information (mean (standard deviation)) with riders and horses classified according to their highest level of competition. Abbreviations: Grand Prix (GP).

	Rider		Horse	
	GP	Small Tour	GP	Small Tour
Number of participants	12	5	9	11
Age	46 (12)	43 (17)	13 (3)	10 (2)
Mass	64.2 (7.4)	65.4 (8.4)		
Height (m)	1.68 (0.07)	1.69 (0.10)	1.65 (0.10)	1.69 (0.05)
Male	1		8	9
Female	11	5	1	2

**Table 2 animals-13-02496-t002:** Rider-level results for each performance measure and judges’ scores for gaits/movements (top left) with ensemble mean (standard deviation (SD)) for each group (bottom left is GP and bottom right is Small Tour) and significance among groups (top right). Means for each gait/movement are separated by color and shown in horizontal bars, as well as SDs as error bars. Gait/movement key is below. Significance was set at *p* < 0.05. Abbreviations: transverse dynamic symmetry (TDS); average deviation in detrended angular impulse (DVar); left (L); right (R).

	GP	Small Tour	
Stride Frequency (Hz)1.3 (0.3)	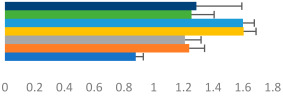	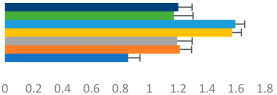	*p* = 0.7171.3 (0.3)
TDS Trunk (m)3.47 (1.8)	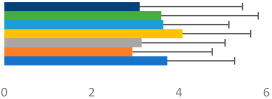	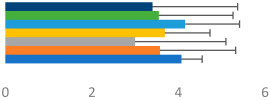	*p* = 0.4813.63 (1.8)
TDS Pelvis (m)1.1 (0.9)	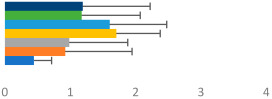	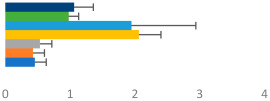	*p* = 0.9231.0 (0.9)
DVar (rad)0.3 (0.2)	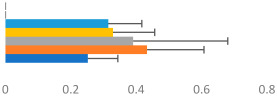	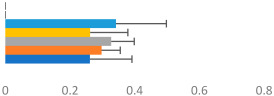	*p* = 0.2500.3 (0.1)
Stability Trunk (ratio)—3.6 (3.2)	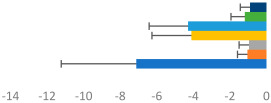	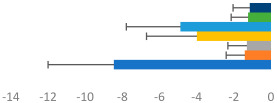	*p* = 0.783—4.0 (3.5)
Stability Pelvis (ratio)—0.3 (0.3)	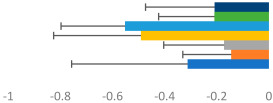	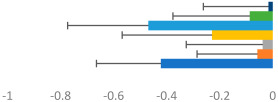	*p* = 0.201—0.2 (0.3)
Judge Gait Score (0–10)6.54 (0.81)	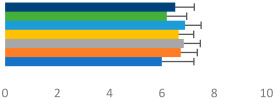	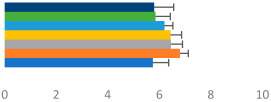	*p* = 0.1936.18 (0.60)
Key: 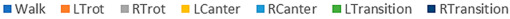

**Table 3 animals-13-02496-t003:** Horse-level results for each performance measure and judges’ scores for gaits/movements (top left) with ensemble mean (standard deviation (SD)) for each group (bottom left is GP and bottom right is Small Tour) and significance between groups (top right). Means for each gait/movement are separated by color and shown in horizontal bars, as well as SDs as error bars. Gait/movement key is below. Significance was set at *p* < 0.05. Abbreviations: transverse dynamic symmetry (TDS); average deviation in detrended angular impulse (DVar); left (L); right (R).

	GP	Small Tour	
Stride Frequency (Hz)1.3 (0.3)	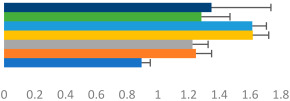	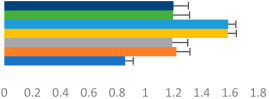	*p* = 0.6781.3 (0.3)
TDS Trunk (m)4.1 (1.9)	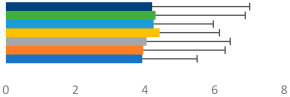	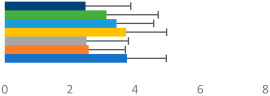	*p* = 0.0943.2 (1.3)
TDS Pelvis (m)1.3 (1.0)	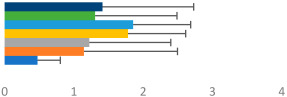	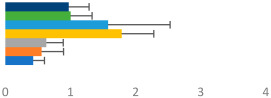	*p* = 0.3041.0 (0.8)
DVar (rad)0.29 (0.10)	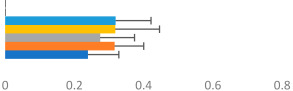	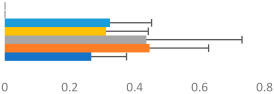	*p* = 0.1480.36 (0.19)
Stability Trunk (ratio)—3.5 (3.4)	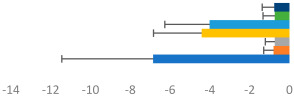	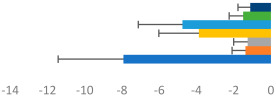	*p* = 0.091—3.8 (3.3)
Stability Pelvis (ratio)—0.25 (0.28)	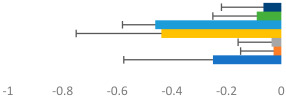	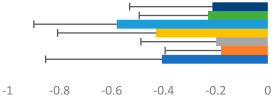	*p* = 0.134—0.36 (0.36)
Judge Gait Score (0–10)6.46 (0.94)	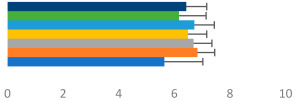	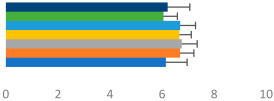	*p* = 0.6356.59 (0.63)
Key: 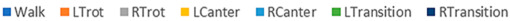

**Table 4 animals-13-02496-t004:** General impression judged scores (mean (standard deviation)) separated by rider and horse levels showing significant differences * (*p* < 0.05) between levels.

	Judged Score	GP	Small Tour	*p*-Value
Rider Level	Position	7.0 (0.4)	6.7 (0.3)	0.124
	Harmony	6.7 (0.5)	6.0 (0.8)	0.027 *
	Aids	6.4 (0.5)	5.5 (0.7)	0.012 *
Horse Level	Position	6.9 (0.5)	7.0 (0.4)	0.618
	Harmony	6.4 (0.9)	6.6 (0.5)	0.681
	Aids	6.1 (0.8)	6.2 (0.7)	0.935

**Table 5 animals-13-02496-t005:** Results of stepwise linear regression analysis for gaits/movements and general impression judged scores. Predictors included stride frequency, trunk transverse dynamic symmetry (TDS trunk), and pelvis stability.

Judged Score	Regression Model and Predictor(s)	R-Value	R^2^	Significance	Durbin–Watson Statistic
Position	(1) TDS Trunk	−0.421	0.178	<0.001	2.213
	(2) TDS Trunk Stability Pelvis	−0.485	0.235	<0.001	
Harmony	(1) TDS Trunk	−0.271	0.074	0.009	2.204
Aids	(1) TDS Trunk	−0.369	0.136	<0.001	1.811
	(2) TDS Trunk Stability Pelvis	−0.460	0.212	<0.001	
Gait Scores	(1) Stride Frequency	−0.252	0.063	0.015	2.096
	(2) Stride Frequency TDS Trunk	−0.361	0.131	0.002	

## Data Availability

Data and templates related to this study are available on request at: https://uclandata.uclan.ac.uk/id/eprint/387.

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
