# Peer review of "Evaluating Overall Performance in High-Level Dressage Horse–Rider Combinations by Comparing Measurements from Inertial Sensors with General Impression Scores Awarded by Judges"

_animals, 2023, doi:10.3390/ani13152496_

Round 1
Reviewer 1 Report
The article presents relevant results and uses consistent evaluation tools. However, some changes in the text may clarify the comparison of the performances of horses judged by different judges. I suggest that the reason why the same two or three judges did not judge all the horses and the impact of this on the performance results is justified since there has been great variability between the results of the evaluations even in FEI competition judges. I suggest reviewing whether the citation of table 4 on line 355 is correct, it seems to me to be referring to table 5.
Author Response
The article presents relevant results and uses consistent evaluation tools. However, some changes in the text may clarify the comparison of the performances of horses judged by different judges. I suggest that the reason why the same two or three judges did not judge all the horses and the impact of this on the performance results is justified since there has been great variability between the results of the evaluations even in FEI competition judges.
We agree that there is expected to be variability between individual judges and chose to assign the pool of judges randomly and in different combinations to the 20 dressage tests. The method of judge assignment and the reasoning behind it has been explained in the methods.
‘A pool of 16 dressage judges who were certified at senior (S) level by the United States Dressage Federation or certified by the Fédération Equestre Internationale were recruited. The judges were randomly assigned to judge the 20 dressage tests with different combinations of judges being used for each test. This method was chosen to distribute high and low scoring judges across the different horse-rider combinations. Each judge scored the performance of the horse and rider in each movement on a scale of 0 to 10 using the same video segments from which the IMU data were extracted. Two or three judges scored each horse-rider combination and their scores were averaged per gait/movement prior to statistical analysis. After completing the test, the riders were judged on three rider scores that represented the components of the general impression score at the end of the FEI test:
- the rider’s position and seat
- the use of discreet and effective aids to influence the horse
- the overall impression of harmony between the rider and the horse’
I suggest reviewing whether the citation of table 4 on line 355 is correct, it seems to me to be referring to table 5.
Thanks for picking up this mistake – citation changed.
Reviewer 2 Report
In the last decade scientific journals covering equine-related research have seen a significant uptick in research focused on the use of IMU sensors specific to horse-rider interaction with particular interest in skill level, and thus, the current manuscript appears to be lacking novelty in it’s current form. Most of the previous publications targeted the dressage industry, similar to the current study, and while it demonstrates a relevance to what is of interest in today’s industry, other than the use of high-level riders, the uniqueness of this study from that of previous work is lost. The title itself should work to separate from other previous work and give better insight as to what is being presented. Due to the current interest in IMU sensors, this would be something to include within the title. Further, the study seems to really focus on comparison between that of judging scores, which separates that from previous work. As such, this information would be of value within the title. While other comparisons within the current study include different levels of riders and different levels of horses, this work has been done previously and may not hold value to further contribution to the industry. The reader needs to clearly see right from the beginning the novelty of this work.
The introduction is lengthy and disjointed at times. It spends much time covering the basics of the dressage sport resulting in the introduction losing focus as to what specifically will be measured. Details concerning the value of IMU sensors and their validity within the scientific community are needed in the introduction. Further discussion of objective predictive performance measures and the shortfall of judges’ scoring systems may help to direct the reader to what is actually being measured within this study. It’s also important to note that nowhere in the title nor the objectives statement is there a mention of the comparison with the judges scores, and yet, that is the focus of the conclusions and what sets this study apart from previous horse-rider interaction studies using IMU sensors. Reconsider objectives statement and hypothesis to reflect a more focused approach considering what sets this study apart from others. As the authors reconsider these statements, include more specific measures as the generalization of “holistic objective measures” is quite vague, and also, include the fact that these are “high-level” dressage riders as that too is unique.
As for methods, while the mention of consent is given by riders is included within this section, was the study protocol evaluated by an institutional review committee for ethical use of both human subjects and animal subjects? Both animals and horses underwent a research protocol unique from their typical work with activities manipulated for the study, and thus, protocol review is required for both humans and animals with protocol numbers given within the methods section.
Further details are needed for lameness evaluations performed on the horses. What protocol was utilized including specifics on diagnostic imaging to ensure that subtle lameness not seen by the human eye within clinical examinations didn’t impact variables measured. Timing of this evaluation and diagnostic imaging should also be given relative to the timing of the study. Similar information concerning the rider is needed as their health status, specifically their soundness, can impact variables measured.
What do the following abbreviations signify: MR, EH, GP, PSG, and Int? Define first and keep in mind that tables and figures should be able to stand alone, thus, any of these abbreviations should be redefined within table/figures. Particularly, since comparisons are between GP and PSG or lower, further details concerning how these individuals and/or horses are classified within these groups must be given reflecting objective measures utilized for these classifications. Specifically, what does “lower” indicate as the goal is to measure “high-level dressage riders”? As such, a more objective and specific term is warranted for “lower” within “PSG or lower”.
What is the “n” for each group? While 17 riders and 20 horses appear to be a part of the study, they are divided into groups, and it appears that some of the groups have extremely low sample populations. For example, appears there was only one rider at PSG or lower that rode a GP horse. Was a power analysis done looking at the needed sample size for each group to achieve statistical significance? How was sample size for each group determined? How was the number of riders, horses, strides, and judges determined? Explain more the grouping briefly discussed in lines 158-159 including how this selection for horse-rider assignment was determined and as to whether this was randomized.
For sensor placement, how was that consistent between each rider and between each horse? Who was responsible for this placement and how was accuracy and consistency ensured? What are qualifications for those assisting with determining proper anatomical placement of humans and horses for proper sensor placement? Explain this “exaggerated movement” within line 171 using more objective, specific terminology.
Video using only 25 Hz for trotting and cantering gaits seems insufficient and even in lines 316-317 the authors mention “video footage was not deemed sufficient”. As such, what is the justification for using 25 Hz and how was video quality and reliability ensured? What was the inclusion/exclusion criteria for gaits utilized within the analysis? Mentions 9 strides, but why would a stride not be included?
Was data collected all on the same day, and if not, how was the same conditions ensured? For riders riding multiple horses was that on the same day and was horse order randomized? Appears that while 2 or 3 judges were used per rider it does not say how the judge was assigned to score horse/rider, and thus, was it randomized and why were some only 2 judges utilized for some and others with 3?
Add in an appendix the dressage test developed for this study as it would be helpful to see the order of gaits and maneuvers performed within the test. Why wasn’t a standard test utilized and who developed this test? Could order of gaits and maneuvers within the test influence potential fatigue of rider and horse within the data collection, and thus, influencing measures?
While the above information needs to be added to the methods section, additional details of the analysis of the data, specifically the IMU sensor data, appears redundant from previous work. The current methods section appears more to be a methodology paper in it’s current format. If a simple reference to some of the previous work in IMU sensors cannot suffice for the methods within this study, then, this manuscript may need to be divided into a methodology manuscript along with another manuscript focused on applying the methodology concerning comparisons made. The additional figures validating methods suggest more a methodology paper, but the authors don’t highlight nor discuss the uniqueness of this methodology. If these methods are not unique, then, streamline the methods utilizing appropriate referencing, and then, expand on the areas discussed above where information is lacking. This point in itself brings up a significant problem with this manuscript that is more obvious within the discussion and conclusions section as these sections again seem disjointed and seem to not reflect the aims of this study nor the title. There’s no mention of comparing upper-level horses in the title nor the aims of the study, and yet, the results and discussion cover differences within the horses, which is not novel from current available publications. Yes, this may be a reflection of the ability of the rider, but the authors fall short of portraying this view. Further, is the focus of this manuscript on how riders influence horse performance as this has been done in previous work. It seems that the judging scores as it relates to the IMU sensor measures of the rider would be of value to the objectives given. The multiple groups of different levels of riders and horses aren’t necessary for the current title, objectives, and conclusions, and depending on the sample size for each group, data from these groups may not be of statistical value. Rewrite discussion to focus on the objectives and title of this study.
See comments above.
Author Response
In the last decade scientific journals covering equine-related research have seen a significant uptick in research focused on the use of IMU sensors specific to horse-rider interaction with particular interest in skill level, and thus, the current manuscript appears to be lacking novelty in it’s current form. Most of the previous publications targeted the dressage industry, similar to the current study, and while it demonstrates a relevance to what is of interest in today’s industry, other than the use of high-level riders, the uniqueness of this study from that of previous work is lost. The title itself should work to separate from other previous work and give better insight as to what is being presented. Due to the current interest in IMU sensors, this would be something to include within the title. Further, the study seems to really focus on comparison between that of judging scores, which separates that from previous work. As such, this information would be of value within the title. While other comparisons within the current study include different levels of riders and different levels of horses, this work has been done previously and may not hold value to further contribution to the industry. The reader needs to clearly see right from the beginning the novelty of this work.
We agree with the reviewer regarding the comparison between objective measurements and the judges’ general impression scores that differentiates this study. The title has been changed to reflect this – thank you for this suggestion.
The introduction is lengthy and disjointed at times. It spends much time covering the basics of the dressage sport resulting in the introduction losing focus as to what specifically will be measured. Details concerning the value of IMU sensors and their validity within the scientific community are needed in the introduction.
We have included a paragraph on the validation and use of IMU systems in horses and horse-rider dyads in lines 121 to 129.
Further discussion of objective predictive performance measures and the shortfall of judges’ scoring systems may help to direct the reader to what is actually being measured within this study. It’s also important to note that nowhere in the title nor the objectives statement is there a mention of the comparison with the judges scores, and yet, that is the focus of the conclusions and what sets this study apart from previous horse-rider interaction studies using IMU sensors.
Reconsider objectives statement and hypothesis to reflect a more focused approach considering what sets this study apart from others. As the authors reconsider these statements, include more specific measures as the generalization of “holistic objective measures” is quite vague, and also, include the fact that these are “high-level” dressage riders as that too is unique.
We have made changes in the hypothesis and emphasized that these are high level riders.
As for methods, while the mention of consent is given by riders is included within this section, was the study protocol evaluated by an institutional review committee for ethical use of both human subjects and animal subjects? Both animals and horses underwent a research protocol unique from their typical work with activities manipulated for the study, and thus, protocol review is required for both humans and animals with protocol numbers given within the methods section.
This journal puts the institutional review information at the end of the manuscript. Human and equine review is from UCLAN and additional equine review from the University of Kentucky. This is the statement:
Institutional Review Board Statement: The study was conducted in accordance with the Declaration of Helsinki and approved by the Institutional Review Board of the UNIVERSITY OF KENTUCKY (Ethics number 2019-3150; 15th October 2019) and the Ethics Committee of the UNIVERSITY OF CENTRAL LANCASHIRE (STEMH 961; 6th February 2019).
Further details are needed for lameness evaluations performed on the horses. What protocol was utilized including specifics on diagnostic imaging to ensure that subtle lameness not seen by the human eye within clinical examinations didn’t impact variables measured. Timing of this evaluation and diagnostic imaging should also be given relative to the timing of the study. Similar information concerning the rider is needed as their health status, specifically their soundness, can impact variables measured.
Symmetry of the horses was checked both subjectively by 2 experienced clinicians and objectively using a validated analysis system based on data from IMUs on the poll and sacrum (Equimoves). This has been included in lines 197-201. See lines 178-180 and 198-200.
For humans included in the study, all participants completed a PAR-Q so their health status was assessed prior to their inclusion in the study.
What do the following abbreviations signify: MR, EH, GP, PSG, and Int? Define first and keep in mind that tables and figures should be able to stand alone, thus, any of these abbreviations should be redefined within table/figures. Particularly, since comparisons are between GP and PSG or lower, further details concerning how these individuals and/or horses are classified within these groups must be given reflecting objective measures utilized for these classifications. Specifically, what does “lower” indicate as the goal is to measure “high-level dressage riders”? As such, a more objective and specific term is warranted for “lower” within “PSG or lower”.
MR and EH are the initials of two of the authors. This has been clarified by preceding these initials with the word authors. Abbreviations for the different levels of competition are now included in the introduction.
What is the “n” for each group? While 17 riders and 20 horses appear to be a part of the study, they are divided into groups, and it appears that some of the groups have extremely low sample populations. For example, appears there was only one rider at PSG or lower that rode a GP horse. Was a power analysis done looking at the needed sample size for each group to achieve statistical significance? How was sample size for each group determined? How was the number of riders, horses, strides, and judges determined? Explain more the grouping briefly discussed in lines 158-159 including how this selection for horse-rider assignment was determined and as to whether this was randomized.
Apologies if this was confusing. We analysed rider level combinations and horse level combinations separately, as there was only one PSG rider who rode a GP horse, so a horse-rider combination analysis was not possible. The groupings were as follows:
For rider level groupings (based on highest competition level): GP riders (n=14) and riders below GP (n=5)
For horse level groupings (based on highest competition level): GP horses (n=8) and horses below GP (n=11)
The description relates to the horse-rider combinations. Riders rode their own horses, so horse/rider combinations were not determined by the research team. We have moved the text and expanded it to clarify this.
This sample size for the study was based on a previous studies. Biau and Barrey, 2004 reported that correlations between subjective and objective scores for experienced horses were largely not significant in a group of 9 horse/rider combinations. Other studies evaluating the rider have often used small numbers of riders. Baxter et al. (2022) identified differences between 8 novice and 8 advanced riders performing shoulder in, but these were measurements from the hip joints (which we were not measuring). As the measurements used in this study were original and we wanted to use high level riders (most of the previous studies did not measure GP/PSG riders) we could not perform a meaningful power calculation.
We used a linear mixed model, as this model is reported to be more robust when analysing unbalanced groups, and we included inferential statistics instead of merely descriptive statistics separated by group levels to provide a reference for future work that may wish to use the methods. To confirm this assumption we conducted another test, comparing 5 GP riders (selected using an Excel random number generator) to the 5 Below GP riders. These were the p-values for each variable:
Stride frequency = 0.892TDS Trunk = 0.659
TDS Pelvis = 0.721
DVar = 0.164
Trunk Stability = 0.684
Pelvis Stability = 0.195
Judged gait score = 0.101
We used nine strides for each gait to ensure that we included only straight-line motion. This is included in section 2.2 procedures.
We have also included more information in relation to how the judges were recruited and how the tests were assigned to them in section 2.2 procedures.
For sensor placement, how was that consistent between each rider and between each horse? Who was responsible for this placement and how was accuracy and consistency ensured? What are qualifications for those assisting with determining proper anatomical placement of humans and horses for proper sensor placement? Explain this “exaggerated movement” within line 171 using more objective, specific terminology.
All researchers involved in the study have many years of experience in applying markers to horses and humans. The person responsible for placing the human markers is a Professor of Equine and Human Locomotion who works in a human sport science department and has taught human biomechanics for over two decades. The people applying the equine markers are all equine veterinarians with PhD degrees in equine gait analysis.
The exaggerated movement refers to swinging the arm through a large of arc of motion prior to tapping the sacral marker to facilitate identifying this event on the videos for synchronization of video and IMU data.
Video using only 25 Hz for trotting and cantering gaits seems insufficient and even in lines 316-317 the authors mention “video footage was not deemed sufficient”. As such, what is the justification for using 25 Hz and how was video quality and reliability ensured? What was the inclusion/exclusion criteria for gaits utilized within the analysis? Mentions 9 strides, but why would a stride not be included?
The videos were used by the judges to analyze the horse-rider performances and to identify the same sequences of IMU data for extraction of variables. None of the variables (except the judges’ scores) were based on video analysis. The camera operator who filmed all the dressage tests at C or A was ill on the day that the video footage of this horse-rider combination was collected. None of the other researchers were as proficient at filming the tests, so the video footage for this horse/rider combination was not included.
Was data collected all on the same day, and if not, how was the same conditions ensured? For riders riding multiple horses was that on the same day and was horse order randomized? Appears that while 2 or 3 judges were used per rider it does not say how the judge was assigned to score horse/rider, and thus, was it randomized and why were some only 2 judges utilized for some and others with 3?
Data was collected from several different barns over a 10 day period. Each horse only performed the test once for the study. For the three riders who rode two horses each, two of the riders rode a different horse on the same day. One of the riders rode a different horse on a different day. The order that they rode the horses in was randomized. We have added more information to clarify this.
Add in an appendix the dressage test developed for this study as it would be helpful to see the order of gaits and maneuvers performed within the test. Why wasn’t a standard test utilized and who developed this test? Could order of gaits and maneuvers within the test influence potential fatigue of rider and horse within the data collection, and thus, influencing measures?
The test was written by one of the authors (HMC). We used a custom test to include all the gaits and movements we were interested in analysing in a broader study. After warming up, the horses performed the trot movements, rested for 5-10 minutes, performed the walk, piaffe and passage movements, rested 5-10 minutes, then performed the canter movements. The duration of the entire test was 8-10 minutes which was well with in the capacity of these fit competition horses.
We have included the test, along with the instructions for judges as a supplementary file.
While the above information needs to be added to the methods section, additional details of the analysis of the data, specifically the IMU sensor data, appears redundant from previous work. The current methods section appears more to be a methodology paper in it’s current format. If a simple reference to some of the previous work in IMU sensors cannot suffice for the methods within this study, then, this manuscript may need to be divided into a methodology manuscript along with another manuscript focused on applying the methodology concerning comparisons made. The additional figures validating methods suggest more a methodology paper, but the authors don’t highlight nor discuss the uniqueness of this methodology. If these methods are not unique, then, streamline the methods utilizing appropriate referencing, and then, expand on the areas discussed above where information is lacking. This point in itself brings up a significant problem with this manuscript that is more obvious within the discussion and conclusions section as these sections again seem disjointed and seem to not reflect the aims of this study nor the title. There’s no mention of comparing upper-level horses in the title nor the aims of the study, and yet, the results and discussion cover differences within the horses, which is not novel from current available publications. Yes, this may be a reflection of the ability of the rider, but the authors fall short of portraying this view. Further, is the focus of this manuscript on how riders influence horse performance as this has been done in previous work. It seems that the judging scores as it relates to the IMU sensor measures of the rider would be of value to the objectives given. The multiple groups of different levels of riders and horses aren’t necessary for the current title, objectives, and conclusions, and depending on the sample size for each group, data from these groups may not be of statistical value. Rewrite discussion to focus on the objectives and title of this study.
We have tried to give a minimum of information describing the location and operating features of the IMUs in this study. The main reason the methodology is rather long is that it contains an explanation of some unique performance variables. Since these are unlikely to be familiar to readers we have illustrated how these variables are calculated to help readers understand their meaning. With respect, we don’t think it would be useful to split this into two manuscripts because the value of the new variables would not be clear without the results section.
We have rewritten the discussion and conclusions which we hope will remedy the concerns raised.
Just to note: We have also provided links to example files for the methods developed in this study.
Reviewer 3 Report
Really interesting study matching objective measures of rider movement to subjective judges’ assessments. I have only minor comments which may be considered to help improve the readability of the paper.
Simple summary – last sentence – Could you be a bit more explicit and say that a higher score was associated with fewer rider asymmetries.
Abstract – As above, could you be more explicit, saying higher scores were associated with greater symmetry measures?
Introduction – I would recommend introducing the dressage competitive levels (for readers unfamiliar with dressage levels) with a brief description here (GP, PSG, Int etc), as they are not defined in the methods.
Methods
Table 1 – Could you clarify if the information presented is mean (SD)? And numbers of participants? Could you divide the information into rider levels (and horse) – as this is how you’ve compared the data in your results. This may also give you further reasoning behind your statement in discussion (line 518-521).
Line 168 – were sensors attached to all 4 hooves of the horse?
Line 189 – presumably the scores were out of 10?
Results
Table 2,3 – is the score for the general impression from the dressage judges? Or one of the 3 components delineated in the methods above?
Line 354 – table 4 doesn’t appear to report results from linear regression. Amend to table 5 (if this is what you’re referring to).
Line 370 – VIF needs definition
Discussion – are the authors planning any future work to match these data to COP under the saddle data? This would be really interesting to look at.
Line 487 – did you measure saddle force data? This reads as if you did.
Line 518-521 – Could you show the information (participant info and age at different levels) by integrating it into Table 1?.
Author Response
Really interesting study matching objective measures of rider movement to subjective judges’ assessments. I have only minor comments which may be considered to help improve the readability of the paper.
Thank you!
Simple summary – last sentence – Could you be a bit more explicit and say that a higher score was associated with fewer rider asymmetries.
Sentence at the end of the simple summary has been extended to include your suggestion.
Abstract – As above, could you be more explicit, saying higher scores were associated with greater symmetry measures?
The last sentence of the abstract has been changed to include this suggestion.
Introduction – I would recommend introducing the dressage competitive levels (for readers unfamiliar with dressage levels) with a brief description here (GP, PSG, Int etc), as they are not defined in the methods.
These have been included in the introduction.
Methods
Table 1 – Could you clarify if the information presented is mean (SD)? And numbers of participants? Could you divide the information into rider levels (and horse) – as this is how you’ve compared the data in your results. This may also give you further reasoning behind your statement in discussion (line 518-521).
We have added text in the table legend to indicate that information is mean and SD, split into the appropriate horse and rider levels and added the numbers of participants.
Line 168 – were sensors attached to all 4 hooves of the horse?
Yes – this has been clarified.
Line 189 – presumably the scores were out of 10?
Yes – this information is now included.
Results
Table 2,3 – is the score for the general impression from the dressage judges? Or one of the 3 components delineated in the methods above?
The ‘Score’ in Tables 2 and 3 are the judges gait scores. We have clarified this in the table and legend.
Line 354 – table 4 doesn’t appear to report results from linear regression. Amend to table 5 (if this is what you’re referring to).
Thank you – this has been amended.
Line 370 – VIF needs definition
We have included a description with the abbreviation in the results as well as the methods for clarification.
‘The variance inflation factor (VIF) values, which detect the severity of multicollinearity, for each model and excluded variables were 1 to 1.723, indicating that collinearity was not evident in the analysis.’
Discussion – are the authors planning any future work to match these data to COP under the saddle data? This would be really interesting to look at.
That’s a really good idea but not planned at present. Might be possible to do retrospectively on an existing data set.
Line 487 – did you measure saddle force data? This reads as if you did.
Added a reference to clarify where the information is from.
Line 518-521 – Could you show the information (participant info and age at different levels) by integrating it into Table 1?.
We have included more information about the riders in Table 1.
Round 2
Reviewer 2 Report
Authors are commended on the revisions that were made, however, additional work is required to move forward with publication of the current work. Although changes in the title allow for further details of what is being done within this study, the authors need to be specific as to what “objective measurements”, meaning clearly state within the title that these are IMU measurements. Also, remove “subjective” from the title and just refer to this as “general impression scores awarded by judges”. Further, while some clarification is given within the introduction as to the direction and scientific background of the study, the authors have, in turn, through the revisions made the introduction too lengthy. Much of the details given concerning dressage can be minimized to focus more on the previous research that has been done staying more objective in the interpretation of the sport and it’s judging system. Potentially, some of the judging criteria and background concerning dressage could be included within the supplementary material such as a table outlining scoring and evaluation procedures as given within judging standards so that the authors can minimize the background information within the introduction. The main focus of the objectives of this study gets lost in trying to give a full synopsis of dressage and the judging of the dressage horse. As any dressage judge would admit, this cannot be done within one paper, particularly within the introduction of one paper, and thus, limit the focus to more the science behind the previous research that is applicable to this study and allow readers that are interested in more details on judging to refer to valuable citations and potentially further explanation given within supplementary materials to explore areas of dressage judging that they are less familiar with. This streamlining of information concerning dressage background will assist in reducing length of introduction and staying true to the science behind this study.
As for the methods, authors did a thorough job of explaining the steps to the project. Do, however, clarify when indicating “below GP” as there are many levels below Grand Prix and with such a small sample size it would be useful to narrow down the specifics of the horse-rider combinations utilized. Instead of using “below GP” give the specific level and/or levels these subjects represent and make sure to do this throughout the manuscript. Also, depending on what levels this “below GP” includes, would this then be considered “high level dressage riders” as stated within the title? Readjust as needed to address this interpretation of what is considered “high level” and ensure within the methodology that it is clearly defined objectively what is the inclusion criteria for being labeled a “high level” dressage rider for this study. Specifically, what levels fall into this categorization of “high level” and is this the same for the horses utilized as the discussion concerning limitations for this study included the explanation of using “less experienced” horses (line 609). Further, do note that while the title indicates just “riders” the evaluation is of horse-rider combination. This too may need further evaluation in reconsidering updates. Again, it is unfortunate that the sample size is so small. Authors give in their response to the reviewer justification for this sample size and what analysis was done to support these low numbers, but this explanation needs to be given within the manuscript within the methodology, and again, a power analysis is the preferred method for statistical determination of suitable sample size. Authors should address accordingly before moving forward with publication.
Finally, similar to the introduction with revisions of the manuscript, the discussion section became too lengthy for what findings are addressed within this study. Authors are recommended to remove lines 571-584 and lines 631-632 as this information does not provide additional impact to the content of the study and this removal of content will assist in reducing the length of this section. Further reduction is recommended.
See previous comments.
Author Response
Authors are commended on the revisions that were made, however, additional work is required to move forward with publication of the current work. Although changes in the title allow for further details of what is being done within this study, the authors need to be specific as to what “objective measurements”, meaning clearly state within the title that these are IMU measurements. Also, remove “subjective” from the title and just refer to this as “general impression scores awarded by judges”.
Thank you for your suggestion. We have revised the title accordingly.
Further, while some clarification is given within the introduction as to the direction and scientific background of the study, the authors have, in turn, through the revisions made the introduction too lengthy. Much of the details given concerning dressage can be minimized to focus more on the previous research that has been done staying more objective in the interpretation of the sport and it’s judging system. Potentially, some of the judging criteria and background concerning dressage could be included within the supplementary material such as a table outlining scoring and evaluation procedures as given within judging standards so that the authors can minimize the background information within the introduction. The main focus of the objectives of this study gets lost in trying to give a full synopsis of dressage and the judging of the dressage horse. As any dressage judge would admit, this cannot be done within one paper, particularly within the introduction of one paper, and thus, limit the focus to more the science behind the previous research that is applicable to this study and allow readers that are interested in more details on judging to refer to valuable citations and potentially further explanation given within supplementary materials to explore areas of dressage judging that they are less familiar with. This streamlining of information concerning dressage background will assist in reducing length of introduction and staying true to the science behind this study.
We have removed a section of the introduction, as requested. The remaining information is important in providing context to justify the aims for the study, so we feel that this should remain.
As for the methods, authors did a thorough job of explaining the steps to the project. Do, however, clarify when indicating “below GP” as there are many levels below Grand Prix and with such a small sample size it would be useful to narrow down the specifics of the horse-rider combinations utilized. Instead of using “below GP” give the specific level and/or levels these subjects represent and make sure to do this throughout the manuscript. Also, depending on what levels this “below GP” includes, would this then be considered “high level dressage riders” as stated within the title? Readjust as needed to address this interpretation of what is considered “high level” and ensure within the methodology that it is clearly defined objectively what is the inclusion criteria for being labeled a “high level” dressage rider for this study. Specifically, what levels fall into this categorization of “high level” and is this the same for the horses utilized as the discussion concerning limitations for this study included the explanation of using “less experienced” horses (line 609). Further, do note that while the title indicates just “riders” the evaluation is of horse-rider combination. This too may need further evaluation in reconsidering updates. Again, it is unfortunate that the sample size is so small. Authors give in their response to the reviewer justification for this sample size and what analysis was done to support these low numbers, but this explanation needs to be given within the manuscript within the methodology, and again, a power analysis is the preferred method for statistical determination of suitable sample size. Authors should address accordingly before moving forward with publication.
We have included a better description of the levels that were tested within the introduction and due to this we have revised below GP to Small Tour, which provides a more accurate description of the lower level riders.
Finally, similar to the introduction with revisions of the manuscript, the discussion section became too lengthy for what findings are addressed within this study. Authors are recommended to remove lines 571-584 and lines 631-632 as this information does not provide additional impact to the content of the study and this removal of content will assist in reducing the length of this section. Further reduction is recommended.
Thank you for your suggestions. We have reduced the discussion further as advised.